# Financial burden of HIV and TB among patients in Ethiopia: a cross-sectional survey

Lelisa Fekadu Assebe ,[1,2] Eyerusalem Kebede Negussie,[2] Abdulrahman Jbaily,[3] Mieraf Taddesse Taddesse Tolla ,[3] Kjell Arne Johansson[1]

[1]Department Of Global Public Health and Primary Care, Faculty of Medicine, University of Bergen, Bergen, Norway
[2]Disease Prevention and Control, Ministry of Health, Addis Ababa, Ethiopia
[3]Department Of Global Health and Population, Harvard T.H.Chan School of Public Health,Harvard University, Boston, Massachusetts, USA

**Correspondence to**
Dr Lelisa Fekadu Assebe; lelfekadu1@gmail.com

## ABSTRACT

**Objectives** HIV and tuberculosis (TB) are major global health threats and can result in household financial hardships. Here, we aim to estimate the household economic burden and the incidence of catastrophic health expenditures (CHE) incurred by HIV and TB care across income quintiles in Ethiopia.

**Design** A cross-sectional survey.

**Setting** 27 health facilities in Afar and Oromia regions for TB, and nationwide household survey for HIV.

**Participants** A total of 1006 and 787 individuals seeking HIV and TB care were enrolled, respectively.

**Outcome measures** The economic burden (ie, direct and indirect cost) of HIV and TB care was estimated. In addition, the CHE incidence and intensity were determined using direct costs exceeding 10% of the household income threshold.

**Results** The mean (SD) age of HIV and TB patient was 40 (10), and 30 (14) years, respectively. The mean (SD) patient cost of HIV was $78 ($170) per year and $115 ($118) per TB episode. Out of the total cost, the direct cost of HIV and TB constituted 69% and 46%, respectively. The mean (SD) indirect cost was $24 ($66) per year for HIV and $63 ($83) per TB episode. The incidence of CHE for HIV was 20%; ranges from 43% in the poorest to 4% in the richest income quintile (p<0.001). Similarly, for TB, the CHE incidence was 40% and ranged between 58% and 20% among the poorest and richest income quintiles, respectively (p<0.001). This figure was higher for drug-resistant TB (62%).

**Conclusions** HIV and TB are causes of substantial economic burden and CHE, inequitably, affecting those in the poorest income quintile. Broadening the health policies to encompass interventions that reduce the high cost of HIV and TB care, particularly for the poor, is urgently needed.

## Strengths and limitations of this study

► This study will be of high value to policies aimed at universal health coverage for HIV and tuberculosis (TB) care in Ethiopia due to the financial risk of seeking care.

► Patient costs from this study can provide empirical bases for national HIV and TB programmes for adopting public finance or health insurance-based financing strategies.

► The patient costs for outpatient and inpatient HIV and TB care are presented together with income and consumption levels of households.

► The cost measurements relied on a patient's ability to remember, which increases risk of recall bias.

► The HIV sampling consisted primarily of urban population and the TB study was limited to specific regions of the country, therefore, these samples are not necessarily nationally representative.

issues still exist.[1–3] Better understanding of factors that affect use of these services would help countries to achieve universal health coverage (UHC) of HIV and TB services.

The population in need for care is still large. Globally, 1.7 million people acquired HIV infection and 10 million new TB cases occurred in 2018. In the same period, more than 2 million people died from HIV and TB.[4 6] In Ethiopia, the prevalence of HIV among adults was 1% (CI: 0.7% to 1.4%) and the incidence rate of TB was 151/100 000 in 2018.[4 6] In order to end HIV and TB, a comprehensive approach should include medical and non-medical interventions such as socio-economic support and poverty alleviation.[7 8]

Although many countries, including Ethiopia, offer 'free' HIV and TB services, the implemented policies do not adequately provide realistic financial risk protection. The health budget in Ethiopia is low ($33.2 per capita) and 31% of overall health financing is out of pocket payments (OOP).[9 10] Hence, patient and their families, often face both

## BACKGROUND

HIV and tuberculosis (TB) are major global health threats that cause a large financial burden on vulnerable populations. Global efforts, in the past two decades, have improved access to lifesaving HIV and TB interventions.[1–3] More than 72 million lives have been saved between 2000 and 2018.[4 5] Nevertheless, high disease burden, inequality in utilisation of healthcare and service quality

**BMJ**

direct and indirect costs, which create financial burden on households.[11 12] A systematic review showed that individuals in low-income countries spend a mean direct cost of $155 per drug susceptible TB and $406 per drug-resistant TB (DR-TB). The productivity losses were two to three times higher than the direct costs of drug susceptible and resistant cases, respectively.[13] Similarly, for HIV, spending ranges from $95 to $2672 in sub-Saharan Africa.[12] Such high costs are related to catastrophic health expenditure (CHE), which occurs when the OOP exceeds 10% of annual income[14 15] or 40% of household non-food expenditure.[16] In addition, OOP expenses for HIV and TB care may crowd out consumption of basic needs and leave vulnerable households in debt/impoverishment.[12 17 18] Furthermore, high levels of patient cost may affect access to care, and lead to poor treatment outcome and prolonged period with infection.[12 14 19–21]

Many factors may lead to CHE; exemptions of HIV and TB services often applies to limited aspects of the basic care package (eg, $CD_4$, viral load, acid-fast bacilli and GeneXpert tests), treatment (eg, antiretroviral therapy (ART), anti-TB drugs). Patients pay for prediagnostic services, ancillary medications, some laboratory testing, imaging, adverse event monitoring, hospitalisation, transportation, food, lodging, etc.[19 22–24] In addition, unavailability of diagnostic services in public health facilities pushes patients to seek care from expensive private providers.[22] Furthermore, low health insurance coverage (around 24% as of 2019), and repeated follow-up visits were important contributing financial risk factors.[25 26] It is also imperative that HIV and TB programmes, needs to monitor household protection from CHE (ie, financial risk protection) and its distribution across income groups (equity), as OOP health spending places greater burdens on the poor.[27 28]

In Ethiopia, few studies have been evaluating the extent of patient cost due to seeking HIV and TB care. The costs related to severe forms of the diseases and assessments across income or consumption groups were lacking from the studies reviewed.[22 23 29] Furthermore, none studied predictors of CHE.[22 23 29] Because HIV and TB are chronic diseases and intimately linked, it is reasonable to look at them jointly. This paper aims to estimate the economic burden and incidence of CHE incurred by standard HIV and TB care across income or consumption quintile among Ethiopian households. Moreover, we will assess factors associated with CHE for HIV and TB.

## METHODS
### Study setting and population
In this study, a nationwide household survey for HIV[10] and a cross-sectional health facility based survey for TB, were used to estimate direct and indirect costs, and CHE.

Data for HIV were collected from mid-September 2016 to mid-October 2016. The total estimated number of people living with HIV (PLHIV) was 722747 in Ethiopia. However, for TB, data were collected from December 2018 to September 2019 in three zones (ie, zone 3 of Afar region, and Jimma and Adama special zones of Oromia region). The three zones were purposely selected and represents 4 million people mirroring the country's geographical and socio-economic heterogeneity. The zones account for 10% and 13% of the TB cases in Oromia and Afar regions (and 6% of the national prevalence), respectively.

### Sample size and sampling technique
For HIV, PLHIV associations were used as a sampling frame to select HIV participants, as there is no national registry of PLHIV. The association operates in major cities in all regions and its members were primarily residents in urban areas. The estimated sample size was 4200. The response rate ranges from 92% to 100% across regions. A two-stage stratified cluster-sampling method was employed. In stage one, a sample of 105 HIV associations from a total of 588 were randomly selected and allocated to each region using probability proportional to the size. In stage two, 40 HIV members from each sampled association, in total 4171, were randomly selected and interviewed. Among the study participants (ie, 4171), 1006 had HIV-related care during the data collection period and were considered in our analysis.

For TB, the sample size was calculated using two-population proportion formula with 80% power, 5% type I error, 95% CI and using 39%[21] of households incur CHE among the richest income quartile; and to detect 15% point difference revealed 186 samples for each quartile. The final sample size with 10% non-response rate was 818 (of which 7% were DR-TB). Systematic random sampling was employed to select 27 public health facilities from the three zones. The total sample size was distributed proportional to the TB case load.[30]

### Patient and public involvement
The research question of this study is in line with the Ethiopian tuberculosis research plan developed through multiple consultative processes involving broader stakeholders including patient representatives. We plan to disseminate the research findings through the national TB research conferences involving researchers, policymakers, stakeholders and affected communities.

### Data collection tools and quality assurance
The data collection was based on a structured questionnaire adapted from the United States Agency for International Development (USAID) for HIV and the World Health Organization (WHO) patient costing tool for TB.[31 32] The questionnaire captures sociodemographic variables, direct cost, indirect cost, productivity loss, assets, income, consumptions and coping-related information. We complemented clinical information for TB through review of medical records. The questionnaire was translated into local languages, pretested and modified accordingly. Trained data collectors under close supervision undertook the face-to-face interview. TB patients

with a minimum of 1 month on treatment were interviewed consecutively and the expenses were reported retrospectively. This schedule of interview is based on WHO's recommendation regarding the cost survey of TB patients.[28 31] Whereas, for HIV, the expenditure in the past 4 weeks for outpatient and 6 months for inpatient care was gathered.

## Patient cost

The costs of seeking standard HIV and TB care were estimated from a patient perspective. The expenditure related to routine care, HIV-related opportunistic infection and managing comorbidities was considered for HIV. Likewise, for TB episode, the expenditure in the pathway to care from onset of symptoms, diagnosis and completion of treatment was included.

Direct costs includes household expenditures for medical (ie, registration/consultation fees, laboratory tests, X-ray, medicines, hospital admission) and non-medical services (ie, special food/nutrition, transportation and guardian cost) net of reimbursement. Indirect costs constitute lost income following the disease episode. In order to estimate the indirect cost, patients were asked to estimate the time lost due to receiving and waiting for care, hospitalisation, transportation, lost working days and guardian time (TB). Then the total time lost was multiplied by an hourly wage rate, which was derived from monthly income/consumption by assuming 22 working days a month and 8 hours a day. For children less than 15 years of age, non-medical direct and indirect TB cost was computed for the guardian. Total patient cost is the sum of all direct and indirect costs.

In order to annualise the cost, an average of four HIV outpatient visits and single TB episode outpatient visits was considered. The frequency of outpatient HIV visits per year was extrapolated on the basis of per capita healthcare visits an individual made over the last 1 month among all PLHIV interviewed. This proportion was annualised to an approximately four visits per patient and year. The TB patient cost was extrapolated for the whole duration of TB episode based on an individual data reported retrospectively. All costs were gathered in local currency (Ethiopian Birr) and converted to US dollar ($) with the 2019 exchange rate of $1=29.1468 Ethiopian Birr.[33] For HIV, the cost was first converted to a reference period (2019) using Ethiopia's consumer price index.[34] Due to unavailability of data, we used household income (for HIV) and consumption aggregates (for TB) as a proxy for the household welfare measure and scaled to per adult equivalence (online supplementary appendix 1). In addition, participants were grouped into five-income/consumption quintiles to reflect the socio-economic strata (online supplementary appendix 1).

An incidence of CHE occurs when direct costs (ie, OOP) exceed the 10% threshold of annual household income/consumption.[14 15 17] In addition to the 10% threshold, we carried out further analysis at 20% threshold of both OOP and total cost, and at 40% of non-food expenditure

to allow for comparison.[16 21 31 35] Furthermore, the distribution of financial burden (measured as ratio of direct/total costs to total household expenditure) across income quintiles was reported using headcount, overshoot and mean positive overshoot[36] (online supplementary appendix 1).

## Data analysis

Data was analysed using Stata V.16 software. The data was summarised using mean with SD or median with IQR due to skewed distribution. The cost was disaggregated by outpatient and inpatient care. A concentration index was used to assess health outcome measure inequality across income quintiles.[36 37] Multivariate logistic regression was conducted to identify determinants of CHE by including significant variables in the univariate analysis. A stepwise regression approach was employed to develop the final model and an adjusted odds ratio (aOR) with 95% CI was reported. P value less than 0.05 declares statistical significance for each test. Multicollinearity was ruled out (variance inflation factor <5). Goodness of fit was checked by Hosmer-Lemeshow test.[38]

## Ethical consideration

Informed written consent was obtained from TB participants. Oromia and Afar Regional Health Bureau, and respective zonal health offices provided permission to undertake the TB study.

## RESULTS

Out of the total 1006 HIV and 787 TB participants, 75% of HIV and 50% of TB were females (table 1). More than two-thirds of the study participants were in the age group between 25 to 44 years for HIV and 1 to 34 years for TB. The median family size was four (IQR, HIV: 3 to 5, IQR, TB: 3 to 6). The mean (SD) household annual income/consumption per adult equivalence was $1188 ($1288) for HIV and $545 ($462) for TB. Seven per cent of TB patients were co-infected with HIV. Almost all (99%) of HIV and 91% of TB/HIV co-infected study participants were receiving ART. About 22% and 6% of HIV and TB patients had both outpatient and inpatient care, respectively. The mean hospital stay was 11 days for HIV and 10 days for TB. The mean (SD) time interval from first healthcare visit to TB diagnosis (health system delay) was 14 (38) days and 85% of TB were diagnosed in public health facilities.

## Patient cost of HIV and TB care

The total mean (SD) patient cost for HIV care was $78 ($170) per year and $115 ($118) for the entire duration of a TB episode (table 2). The mean (SD) direct cost was $54 ($144) for HIV and $53 ($59) for TB, which constitutes 69% and 46% of the total cost, respectively. Medical costs contributed to 68% and 38% of the direct costs for HIV and TB, respectively. Diagnostics and medicine account for 39% of the total HIV cost. The mean (SD)

**Table 1** Socio-demographic and clinical characteristics of HIV and TB study participants (Ethiopia)

| Background characteristics | HIV (n=1006) N (%) | TB (n=787) N (%) |
|---|---|---|
| Gender | | |
| Female | 756 (75) | 396 (50) |
| Age in years (mean, SD) | 40 (10) | 30 (14) |
| Age group | | |
| <18 | – | 110 (14) |
| 18–24 | 14 (1) | 190 (24) |
| 25–34 | 268 (27) | 229 (29) |
| 35–44 | 435 (43) | 117 (15) |
| 45–54 | 202 (20) | 86 (11) |
| 55–64 | 58 (6) | 32 (4) |
| 65+ | 29 (3) | 23 (3) |
| Marital status | | |
| Single | 38 (4) | 309 (39) |
| Married/living together | 493 (49) | 409 (52) |
| Widowed | 277 (27) | 24 (3) |
| Divorced | 137 (14) | 33 (4) |
| Separated | 61 (6) | 11 (2) |
| Place of residence | | |
| Urban | 930 (92) | 394 (50) |
| Rural | 76 (8) | 393 (50) |
| Highest level of education | | |
| Illiterate | 290 (29) | 257 (33) |
| Elementary | 457 (45) | 334 (42) |
| Secondary and higher | 259 (26) | 195 (25) |
| Family size | | |
| ≤4 | 640 (64) | 443 (56) |
| >4 | 366 (36) | 344 (44) |
| Annual household income/consumption | | |
| Lowest | 200 (20) | 159 (20) |
| Second | 470 (21) | 298 (20) |
| Middle | 797 (18) | 443 (20) |
| Fourth | 1342 (20) | 631 (20) |
| Highest | 3084 (21) | 1198 (20) |
| ART status | | |
| On ART | 996 (99) | 52 (91) |
| Not on ART | 10 (1) | 5 (9) |
| Past history of illness* | | |
| Yes | 1006 (24) | 61 (8) |
| No | 3165 (76) | 726 (92) |
| Type of visit | | |
| Outpatient | 790 (79) | 739 (94) |

Continued

**Table 1** Continued

| Background characteristics | HIV (n=1006) N (%) | TB (n=787) N (%) |
|---|---|---|
| Inpatient | 216 (22) | 47 (6) |
| Number of visits per year/TB episode | | |
| Outpatient | 4428 (4 visits/patient) | 48 720 (70 visits/patient)† |
| Inpatient | 249 (1 visit/patient) | 2409 (73 visits/patient)† |
| Type of TB | | |
| Pulmonary TB | – | 507 (65) |
| Extra-pulmonary TB | – | 222 (28) |
| Drug-resistant TB | – | 57 (7) |

*HIV-related comorbidities (HIV), and history of TB (TB).
†The number of total visits per patient reaches 125 for outpatient and 135 for inpatient drug-resistant TB cases.
ART, antiretroviral therapy; TB, tuberculosis.

indirect cost was $24 ($66) for HIV per year, and $63 ($83) per TB episode. The productivity loss related to TB follow-up visits accounts for 36% of the total cost.

The mean (SD) total cost was $63 ($165) for annual outpatient HIV care visit and $110 ($114) for the whole duration of outpatient TB care (table 2). Similarly, the mean (SD) total cost for each hospitalisation was $96 ($139) for HIV and $105 ($78) for TB. For patients having both outpatient and inpatient visits, the mean cost reaches $133 ($178) (p<0.001) for HIV and $ 217 ($157) (p<0.001) for TB.

For TB, the patient costs incurred prior to initiation of treatment are equal to the cost from initiation to completion of TB treatment (paired t-test >0.05). The mean (SD) total cost for those with TB/HIV co-infection reached $188 ($33). Similarly, the total cost of care significantly varies by type of TB, it was $104 ($107) for pulmonary, $140 ($138) for extra-pulmonary and $446 ($732) for DR-TB (Kruskal-Wallis test 41.1, p<0.001) (online supplementary appendix table A1).

### Coping costs

HIV and TB care results in adverse financial consequences for households. Nearly 24% of HIV and 68% of TB study participants have adopted coping mechanisms. Nine per cent of HIV and 4% of TB patients have borrowed money; while, 2% of HIV and 19% of TB patients sold their household assets. Furthermore, 12% of HIV patients relied on family assistance and 16% of TB patients used their savings to cope with the costs. Only 2% of HIV and 6% of TB participants are covered by health insurance.

As shown in figure 1, the spikes in the Pen's parade graph revealed that the healthcare costs of HIV and TB cause a large decrease in annual income/consumption for many of the households. The consumption drop for

**Table 2** Distribution of household direct, indirect and total cost of HIV and TB care across main cost category in Ethiopia (expressed in $)

| Cost category ($) | HIV (n=1006) | | | TB (n=729) | | |
|---|---|---|---|---|---|---|
| | Outpatient (per year) | Inpatient (per single visit) | Total | Outpatient (per TB episode) | Inpatient (per single visit) | Total |
| **(I) Direct medical cost** | | | | | | |
| Consultation fee | | | | | | |
| Mean (SD) | 6 (15) | 1 (5) | 6 (15) | 2 (5) | 11 (16) | 3 (7) |
| Median (IQR) | 0 (0–6) | 0 (0–1) | 0 (0–5) | 1 (0–2) | 7 (1–14) | 1 (0–2) |
| Investigation cost | | | | | | |
| Mean (SD) | 14 (45) | 9 (19) | 15 (47) | 8 (13) | 14 (16) | 9 (15) |
| Median (IQR) | 0 (0–11) | 0 (0–9) | 1 (0–12) | 4 (0–12) | 7 (3–18) | 5 (0–13) |
| Drug cost* | | | | | | |
| Mean (SD) | 12 (40) | 22 (63) | 16 (48) | 7 (14) | 16 (21) | 8 (16) |
| Median (IQR) | 0 (0–10) | 4 (0–19) | 1 (0–13) | 1 (0–10) | 8 (3–20) | 2 (0–10) |
| Subtotal | | | | | | |
| Mean (SD) | 32 (88) | 32 (77) | 37 (98) | 17 (24) | 40 (37) | 20 (30) |
| Median (IQR) | 4 (0–34) | 8 (0–32) | 6 (0–38) | 10 (1–22) | 30 (16–45) | 11 (1–25) |
| **(II) Direct non-medical cost** | | | | | | |
| Transportation fee | | | | | | |
| Mean (SD) | 11 (55) | 19 (34) | 13 (53) | 8 (10) | 7 (7) | 8 (10) |
| Median (IQR) | 2 (0–8) | 6 (0–20) | 3 (0–9) | 5 (2–11) | 7 (3–9) | 5 (2–11) |
| Food/accommodation | | | | | | |
| Mean (SD) | 3 (50) | 10 (30) | 5 (48) | 23 (29) | 28 (34) | 25 (31) |
| Median (IQR) | 0 | 0 (0–5) | 0 | 15 (7–27) | 16 (7–38) | 16 (7–29) |
| Subtotal | | | | | | |
| Mean (SD) | 14 (80) | 29 (49) | 17 (76) | 31 (34) | 35 (36) | 33 (37) |
| Median (IQR) | 2 (0–8) | 10 (0–38) | 3 (0–10) | 21 (11–38) | 20 (9–44) | 21 (12–40) |
| Total direct cost | | | | | | |
| Mean (SD) | 46 (142) | 60 (113) | 54 (144) | 48 (50) | 75 (68) | 53 (59) |
| Median (IQR) | 12 (1–45) | 22 (4–74) | 15 (2–57) | 35 (17–61) | 54 (26–94) | 36 (18–64) |
| **(III) Indirect cost** | | | | | | |
| Foregone income before treatment† | | | | | | |
| Mean (SD) | 0 | 0 | 0 | 11 (28) | 0 | 11 (26) |
| Median (IQR) | 0 | 0 | 0 | 0 (0–5) | 0 | 0 (0–3) |
| Foregone income during treatment | | | | | | |
| Mean (SD) | 11 (44) | 22 (41) | 16 (50) | 11 (26) | 12 (31) | 10 (26) |
| Median (IQR) | 0 | 8 (3–19) | 0 (0–8) | 0 (0–4) | 0 | 0 (0–3) |
| Time loss related cost | | | | | | |
| Mean (SD) | 6 (19) | 14 (52) | 8 (30) | 42 (59) | 18 (24) | 42 (58) |
| Median (IQR) | 2 (1–4) | 2 (1–7) | 2 (1–5) | 27 (14–49) | 8 (5–22) | 27 (14–49) |
| Subtotal | | | | | | |
| Mean (SD) | 17 (54) | 35 (75) | 24 (66) | 62 (84) | 30 (37) | 63 (83) |
| Median (IQR) | 3 (1–8) | 13 (5–32) | 4 (2–17) | 36 (17–77) | 9 (5–39) | 36 (17–78) |
| **(IV) Total cost** | | | | | | |
| Mean (SD) | 63 (165) | 96 (139) | 78 (170) | 110 (114) | 105 (78) | 115 (118) |
| Median (IQR) | 20 (5–63) | 52 (25–109) | 27 (7–80) | 79 (46–140) | 87 (39–158) | 81 (47–150) |

*Drug other than anti-retroviral and anti-TB drugs.
†Not captured in HIV survey.

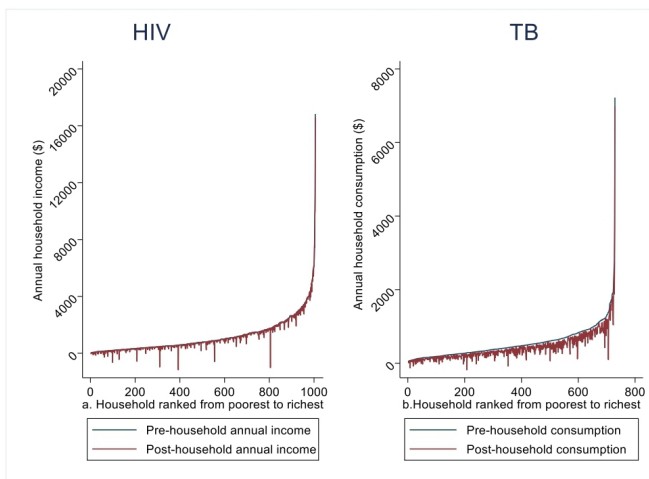

**Figure 1** Pen's parade of household annual income/consumption gross and net of payments for HIV (a) and TB care (b) (Ethiopia). TB, tuberculosis.

TB is more pronounced than the income drop for HIV (figure 1).

As shown in table 3, there are inequalities in OOP costs among HIV and TB participants. The total cost rises steadily across the income quintiles and is concentrated among the richer quintiles (ie, the richer the quintile, the higher the cost). From the lowest to the highest income/consumption quintile, the mean (median) total cost of HIV increases from $53 ($20) to $133 ($60), with significant difference among income quintiles (Kruskal-Wallis test 44.7, p<0.001), and for TB the cost increases from $50 ($36) to $202 ($148) (Kruskal-Wallis test 206.5, p<0.001). In general, the median cost (both direct and indirect)

seems to be a bit higher for TB patients as compared to that of HIV patients.

### Incidence and intensity of CHE

At the 10% threshold, the overall CHE incidence of HIV was 20% (197 households); with 43% of the poorest and 4% of the richest household experiencing CHE ($\chi^2$ for trend −10.58, p<0.001). The incidence is 33% for individuals with inpatient HIV care. The corresponding level of TB was 40% (291 households); with 58% and 20% of the poorest and richest income quintile experienced CHE, respectively ($\chi^2$ for trend −6.79, p<0.001) (table 4). The incidence was much higher for those with TB/HIV co-infection (48%), DR-TB (62%) and was almost universal (94%) for hospitalised TB patients. At the 20% threshold of total expenses, 48% (353 households) of TB households experienced catastrophic total costs (figure 2) (online supplementary appendix table A2).

In our study, for example, the mean overshoot of TB-related CHE was 6.3% (range: 1.9% to 15.3%). On average, households spent 6.3% beyond the 10% threshold for TB care. However, the average positive overshoot among households that experienced CHE was 15.8% (range: 9.2% to 26.6%). Thus, on average, households that experienced CHE spent 25.8% (10% threshold+mean positive overshoot) of their total annual consumption for TB care (figure 2) (online supplementary appendix table A2).

### Inequality in financial risk

In addition, as shown in figure 3, inequality in financial risk across income/consumption groups exists. The concentration curves for HIV and TB care costs lie

**Table 3** Mean (median) HIV and TB patient costs per year across income quintiles in Ethiopia (expressed in $)

| Disease category | Income quintiles | Cost type | | | | | |
| | | Direct | | Indirect | | Total cost | |
| | | Mean (SD) | Median (IQR) | Mean (SD) | Median (IQR) | Mean (SD) | Median (IQR) |
|---|---|---|---|---|---|---|---|
| HIV | Poorest | 45 (92) | 14 (2–48) | 8 (15) | 2 (1–7) | 53 (97) | 20 (3–61) |
| | Poor | 55 (170) | 13 (0–52) | 10 (19) | 3 (1–9) | 65 (173) | 22 (4–65) |
| | Middle | 50 (118) | 19 (3–49) | 16 (30) | 3 (2–16) | 66 (136) | 32 (7–65) |
| | Rich | 47 (89) | 15 (3–57) | 25 (51) | 4 (3–20) | 71 (111) | 27 (7–81) |
| | Richest | 73 (206) | 19 (3–79) | 60 (124) | 14 (5–59) | 133 (262) | 60 (12–132) |
| P value* | | 0.358 | | <0.001 | | <0.001 | |
| TB | Poorest | 31 (35) | 23 (7–41) | 19 (20) | 11 (6–24) | 50 (44) | 36 (20–66) |
| | Poor | 44 (48) | 27 (15–54) | 36 (32) | 27 (13–45) | 79 (65) | 60 (38–96) |
| | Middle | 49 (53) | 36 (17–61) | 57 (46) | 40 (22–82) | 106 (79) | 82 (57–137) |
| | Rich | 61 (48) | 49 (28–84) | 79 (63) | 58 (27–117) | 140 (96) | 118 (68–181) |
| | Richest | 78 (86) | 54 (34–99) | 124 (145) | 70 (40–163) | 202 (189) | 148 (88–260) |
| P value* | | <0.001 | | <0.001 | | <0.001 | |

*Kruskal-Wallis test.
TB, tuberculosis.

**Table 4** Multivariate logistic regression model of determinants of CHE for TB and HIV care at a 10% threshold of household income/consumption (Ethiopia)

| Variable | aOR (95% CI) | P value |
|---|---|---|
| **TB** | | |
| Frequency of visits* | 2.4 (1.9 to 3.1) | <0.001 |
| **Hospitalisation** | | |
| No | Ref. | |
| Yes | 30.6 (4.8 to 199.8) | 0.001 |
| **Income quintiles** | | |
| Richest | Ref. | |
| Rich† | 4.1 (2.1 to 7.8) | <0.001 |
| Middle | 4.9 (2.5 to 9.4) | <0.001 |
| Poor | 7.0 (3.6 to13.7) | <0.001 |
| Poorest | 14.6 (7.5 to 28.3) | <0.001 |
| **Place of diagnosis** | | |
| Government | Ref. | |
| Private | 2.6 (1.5 to 4.3) | <0.001 |
| **TB/HIV co-infection** | | |
| No | Ref. | |
| Yes | 3.2 (1.6 to 6.2) | 0.001 |
| **Insurance (ie, CBHI)** | | |
| Yes | Ref. | |
| No | 2.7 (1.1 to 6.7) | 0.038 |
| **Type of TB** | | |
| Bacteriologically-confirmed TB | Ref. | |
| Clinically-diagnosed TB | 1.6 (1.0 to 2.8) | 0.075 |
| Extra-pulmonary TB† | 2.6 (1.8 to 4.0) | <0.001 |
| **HIV** | | |
| Frequency of visits per year* | 1.07 (1.003 to 1.1) | 0.04 |
| **Hospitalisation** | | |
| No | Ref. | |
| Yes | 3.3 (2.2 to 4.9) | <0.001 |
| **Income quintiles** | | |
| Richest | Ref. | |
| Rich | 2.5 (1.1 to 5.8) | 0.025 |
| Middle | 4.5 (2.1 to 9.8) | <0.001 |
| Poor | 9.4 (4.5 to 19.5) | <0.001 |
| Poorest | 18.4 (8.9 to 37.7) | <0.001 |

*Variable treated as continuous.
†Overall test is significant.
aOR, adjusted OR; CBHI, community-based health insurance; CHE, catastrophic health expenditures; TB, tuberculosis.

below the 45° line of equality, which shows a greater concentration of the costs among the rich. However, the financial burden is higher among the poor—the concentration curves for HIV and TB care expenditure in relation to income/consumption lie above the 45° line of equality.

### Determinants of CHE

In the multivariate analysis (table 4), three variables were independently associated with HIV related CHE: hospitalised patients (aOR: 3.3, 95% CI: 2.2 to 4.9), being poorest (aOR: 18.4, 95% CI: 8.9 to 37.7) and poor (aOR: 9.4, 95% CI: 4.5 to 19.5) were associated with catastrophic HIV care expenditures. Moreover, every additional visit for HIV care increases the odds of CHE by 7% (aOR: 1.07, 95% CI: 1.003 to 1.1).

Seven variables were significantly associated with TB-related CHE (table 4): private facility diagnosis (aOR: 2.6, 95% CI: 1.52 to 4.33), extra-pulmonary TB (aOR: 2.6, 95% CI: 1.77 to 3.95), hospitalised patients (aOR: 30.6, 95% CI: 4.77 to 199.83), being poorest (aOR: 14.6, 95% CI: 7.49 to 28.26) and TB/HIV co-infection (aOR: 3.2, 95% CI: 1.63 to 6.15) were very likely to have TB-related CHE as compared with their counterparts after adjusting for other variables. Every additional visit for TB diagnosis increases the odds of experiencing CHE by 2.4 times (aOR: 2.4, 95% CI: 1.92 to 3.05). Households with a health insurance scheme have protection from CHE (aOR 2.7; 95% CI 1.06 to 6.73).

### DISCUSSION

In this study, we tried to estimate the OOP, total cost, incidence and determinants of CHE among individuals seeking HIV and TB care. This could provide valuable insight into the level of financial risk protection that a health system offers to its population.

### HIV care costs and CHE

In Ethiopia, where ART is given free-of-charge, PLHIV had to pay a total of $78 per year for HIV care ($19.5 per visit). The total HIV patient cost was equivalent to 7% of their annual income. As HIV requires lifelong care, such costs have a devastating impact on affected households. Our estimate was lower than the level and rate found in Cameroon 17%,[39] Ethiopia 21%,[23] Nepal 28.5%[40] and South Africa (30%).[41] These variations might arise from different study settings;[39 40] high expenses of additional food and time loss.[41] Similarly, more centralised HIV service delivery, delays in seeking care and long distance travel to access services may explain the difference with the previous study from Ethiopia. Additionally, the annual income in this study was twice that of the latter study.[23] However, a study on outpatient HIV care in Nigeria reported one-fourth of the cost in this study ($21.76).[42]

In our study, the average direct OOP expenditure for HIV care was $54 (ie, 69% of the total cost) and comparable with that of the previous studies from Lao, Ethiopia and Nepal.[19 23 40] Diagnostics, medicines and transportation costs constitute the largest share, and may pose a serious challenge to the success of the HIV programme. This calls for public financing policies (ie, free of charge diagnosis and treatment of HIV-related comorbidities for vulnerable groups) in the next steps of UHC expansion. The productivity loss found here was one-half of

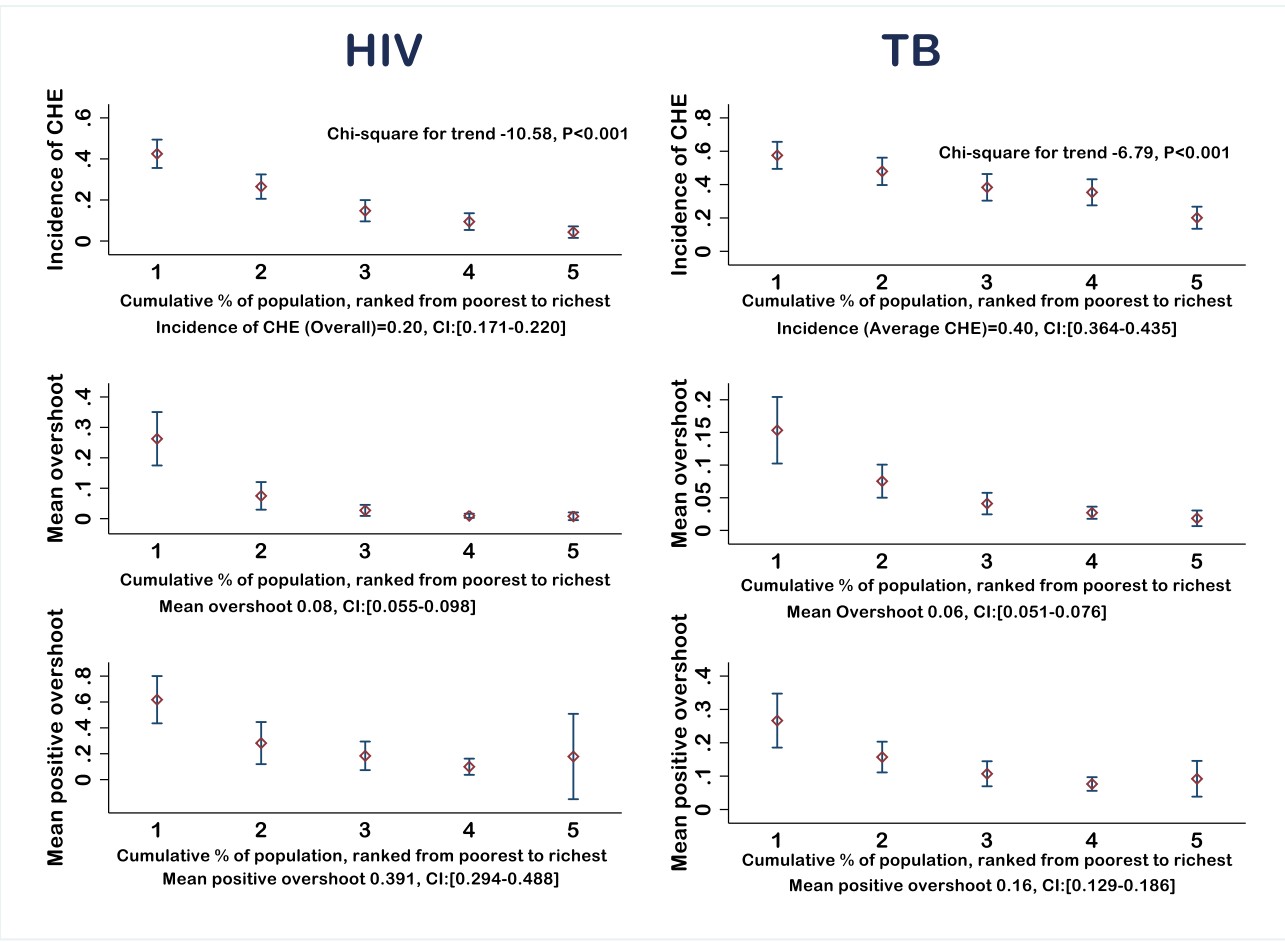

**Figure 2** Interval plot (with 95% CI for the mean) of incidence and intensity of HIV and TB-related catastrophic health expenditures (CHE) across income quintiles, using 10% threshold (Ethiopia). CI, confidence interval, TB, tuberculosis.

that of the previous studies in Ethiopia and Nepal,[23 40] but one-fourth of that of Lao study.[19] The difference in productivity losses with previous Ethiopian study could be attributed to a more centralised provision of HIV services and backlog of patients with advanced HIV diseases, unlike this study. The total costs of HIV care increases as income rises. The equity ratios (Q1:Q5, 0.39) showed higher expenditure among the richest income quintile, which is consistent with a study from southeast Nigeria.[43]

In our study, about a fifth (20%) of patients seeking HIV care experienced CHE. A study from India also depicts similar findings.[44] However, the rate is lower than that of a study from Cameroon.[39] Furthermore, the incidence of HIV-related CHE remained relatively high; in particular, where poorest households suffer more. Similar socio-economic inequality was observed in previous studies,[42 43] highlighting the importance of rendering equitable access to all in need of HIV care, particularly for

the poor.[39] Consistent with previous studies, being poor is associated with higher CHE.[19 45] In addition, the poor were pushed further beyond the CHE threshold than the better off. This is of great concern, as health shocks are slightly managed by the poor through reduction of basic requirements to compensate for HIV care. Similar to previous studies, being hospitalised was a stronger determinant of CHE.[19 42]

### TB care costs and CHE

In our study, patients with TB incurred a total cost of $115 per episode and represents 21% of the annual household income, comparable to a study in South Africa (22%).[46] However, the total cost was less than the figure reported by previous studies in Ethiopia,[22 29] but twice higher than the finding from southern Ethiopia.[47] The variation in cost from previous studies arises from high expenditure on nutritional supplements ($72 vs $25),[29] lower level of

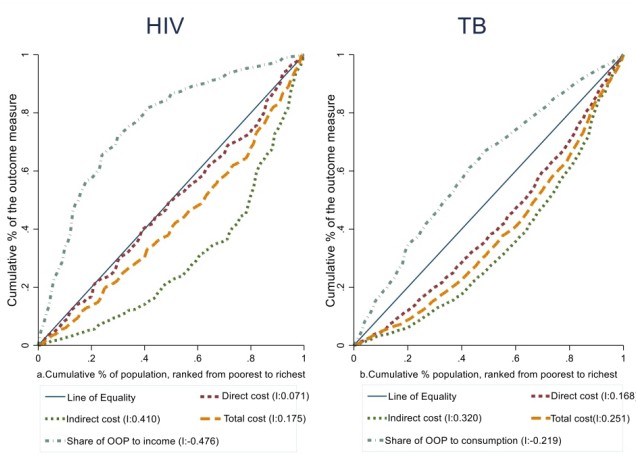

**Figure 3** Concentrations curves and index (I) for direct, indirect, total cost and share of OOP to income/consumption for (a) HIV and (b) TB services in Ethiopia. OOP,out-of-pocket payment; TB, tuberculosis.

seeking diagnostic care from public health facilities (64% vs 85%), high rates of clinically diagnosed cases (49% vs 24%), diagnostic delays result in longer losses of working days,[22] only direct cost captured.[47] The average patient costs of DR-TB care were four times higher than drug-susceptible TB care. However, the DR-TB cost was three times lower than findings from another Ethiopian study.[48] Still the devastating nature of DR-TB may put patients at special risk of CHE. The lower costs is mainly due to decentralisation of DR-TB services in recent years and the introduction of shorter multidrug-resistant TB treatment regimen (ie, 71% the study participants took 9-month to 12-month regimen).

In this study, one-half of the mean TB cost was incurred prior to initiation of treatment, which was consistent with many previous studies.[11 21 22 49 50] This is mostly due to payment while demanding for proper diagnosis of TB.[21] In addition to the financial burden of high pretreatment costs, it will be a barrier to complete the diagnostic process, and to timely access treatment and care. These emphasise the need for early case finding with rapid and point of care TB diagnostics, involvement of private care providers and instituting effective referral and linkage.[47]

The direct patient costs incurred constitute 46% of the total, comparable with findings from elsewhere (36%)[51] and systematic review results (40%).[11] However, the proportion was higher than the finding from southwestern Ethiopia (29%) and lower than the report from central Ethiopia (71%).[22 29] Consistent with previous studies, non-medical and indirect costs represent a large share of the TB cost, while medical cost represent less than 20%.[11 22 41] Therefore, ensuring the expansion of TB service package through the effective integration of a health insurance scheme and decentralisation of services can reduce the direct costs.

A higher percentage of households incurs CHE for TB care (40%), which was comparable with studies from Fiji (40%), Ghana (47.6%), China (53%), Philippines (35%) and lower than reported rates from Ethiopia (63%), Nigeria (65%) and Benin (72%).[29 49 50 52–54] However, CHE

for TB care was higher in this study compared with findings in studies from India (21%) and Malaysia (6%).[51 55] This variation might arise from cost estimation method, study setting, health system and socio-economic differences. The TB-related CHE was higher than the reported rate for HIV. The main reasons for this are the differences in treatment duration, follow-up frequency and care access between HIV and TB. TB patients experience a very onerous set of direct and indirect costs during diagnostic and intensive phase of directly observed short course therapy (ie, more intense for retreatment, extra-pulmonary and DR-TB cases). After treatment completion and possible sputum conversion, TB patients are less likely to face additional costs. Even though the annual cost of HIV care is lower, PLHIV faces these costs over its lifetime because HIV infection is a chronic disease that needs lifelong treatment. The comparison of incidence and intensity of CHE for TB and HIV is also complicated by the use of income for HIV and consumption for TB-related computations. In developing countries, income is a poor self-reported estimator of welfare due to more common informal employment, seasonal agricultural activities and widespread reluctance to disclose income.[15] Therefore, income could understate the welfare of the household, whereas using consumption may overstate the condition of the household because of the use of dissaving/borrowing to smooth consumption over time.[56]

The mean overshoot for TB was 6.3% and the mean positive overshoot was 15.3%, both were similar with finding from Benin (7.8% and 14.8%) and Nigeria (6.0% and 9.3%).[50 52] TB, inequitably, imposes a greater burden of CHE on the poor households. Even though poor households tend to spend less, a higher share of their income is spent on seeking TB care.[12 29 46 50 52] In addition, the excess CHE beyond the threshold was inequitably high among the poor. This finding is also in line with other studies.[29 50 52] Similar to other studies, hospitalisation, income status and TB/HIV coinfection were among the key determinants of TB-related CHE.[28 49 50 52 55] In addition, even if the health insurance coverage (ie, community-based health insurance) was low and is limited to medical costs, we found protective effect of the scheme against CHE.[57] However, health insurance per se does not alleviate the major TB costs.

## Study limitation and strengths

This study has some limitations. First, the cost measurements relied on patient's ability to remember, which increases risk of recall bias. However, we reduced the recall bias by interviewing participants who sought HIV and TB care within the past 1 month, when patients have a better recollection of their pathway to care.[28 31] Second, HIV costs may be overestimated when aggregated over a 1-year period. Third, our findings may not be representative of all patients with TB in Ethiopia, as the study is limited to specific regions of the country. Similarly, the PLHIV associations operate in high HIV prevalence urban areas, where members of these associations may not be representative of both rural and non-members. Moreover, undetected HIV and TB cases not seeking care were not addressed. Despite

these limitations, we used a standard tool and method to conduct the study. We believe that our study will be of high value to inform policy, at national and subnational levels, related to financial risk protection of both diseases, which is central in achieving UHC.

## Policy implications

Despite OOP exemption of HIV and TB services in Ethiopia, we found that there is a large gap between the actual level of financial protection provided and the ideal goal. Our findings highlight important policy implications. First, more patient-centred care with effective diagnostics, appointment spacing for stable patients and community-based treatment are required to improve the delivery of HIV and TB services. Second, strategies are required to ensure social and financial risk protection for the households affected by HIV and TB. This requires effective integration of HIV and TB services with social and financial protection schemes, including the provision of travel vouchers, nutritional support and paid sick leaves through multisectoral collaboration.

## CONCLUSION

HIV and TB affected individuals and their households in Ethiopia face substantial costs in seeking care despite 'free medical services'. The incidence of CHE related to HIV and TB care was high in all income quintiles, though more so in the poorest households. Policymakers should introduce patient-centred care; expand social and financial risk protection measures to minimise the high patient cost of HIV and TB care, particularly among vulnerable populations.

**Acknowledgements** The authors would like to thank all HIV and TB patients and supervisors who participated in this study. We thank all stakeholders; the Ministry of Health, Ethiopia; Ethiopian Public Health Institute; Oromia and Afar Regional Health Bureau with their respective districts for making this research possible.

**Contributors** LFA, KAJ, MTT: conception and design of the study. LFA: acquisition of data, data analysis and drafted the paper. LFA, AJ, KAJ, MTT, EN: critically reviewed the paper and gave approval for the final version to be published.

**Funding** Bill & Melinda Gates Foundation, grant number OPP1162384, supported this study.

**Competing interests** None declared.

**Patient and public involvement** Patients and/or the public were involved in the design, or conduct, or reporting or dissemination plans of this research. Refer to the Methods section for further details.

**Patient consent for publication** Not required.

**Ethics approval** The study was approved by medical and health research ethics in Norway (2018/1647/REK) and Ethiopian Public Health Institute (EPHI-IRB-121–2018).

**Provenance and peer review** Not commissioned; externally peer reviewed.

**Data availability statement** Data are available upon reasonable request. The data supporting the conclusion of this study can be available upon reasonable request from the corresponding author.

**ORCID iDs**
Lelisa Fekadu Assebe http://orcid.org/0000-0002-7857-3349
Mieraf Taddesse Taddesse Tolla http://orcid.org/0000-0002-9832-7807

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
