## [Reviewer comments · BMJ Open]

ARTICLE DETAILS

TITLE (PROVISIONAL)	The financial burden of HIV and TB among patients in Ethiopia: a cross-sectional survey
AUTHORS	Assebe, Lelisa; Negussie, Eyerusalem; Jbaily, Abdulrahman; Tolla, Mieraf Tadesse; Johansson, Kjell Arne

VERSION 1 – REVIEW

REVIEWER	Adebisi Yusuff Adebayo Global Health Focus Africa (Nigeria)
REVIEW RETURNED	29-Jan-2020

GENERAL COMMENTS	Thank you for this interesting article. This article is well-written and scientifically sound. However, the reference style and in-text citations format of the manuscript is not in accordance to the journal's instruction for authors. Please revise - go through an already published article to see how it should be structured.
---

REVIEWER	Arin Dutta Palladium, USA
REVIEW RETURNED	07-Feb-2020

GENERAL COMMENTS	The authors have added usefully to the literature on the economic burden of accessing care for HIV and TB in sub-Saharan Africa. There are some matters which could benefit from a revision. MINOR COMMENT ----- The stated rationale and methods for the study could be refined. The case for catastrophic health expenditures (CHE) is made very strongly, perhaps too strongly, and uses data from other settings than Ethiopia. The range of costs “per TB episode” stated is very high and from a systematic review with studies conducted prior to 2005; and the relevance to the situation in Ethiopia in late 2019 could be more carefully posed, where TB care is an “exempted service” and first- and second-line drugs are available for free based on external grant funding. MAJOR COMMENTS ----- Specific comments: Page 6 line 36-37: clarify “sample of 105 HIV associations”. What type of sampling units are these? With PPS sampling, how did the authors ensure that areas with important characteristics but smaller “HIV associations” were included? Page 6 line 48-49, for sampling, the point estimate of the share of households that may incur CHE in the top income quartile is from
---

	a study in Peru. Its suitability to Ethiopia as a basis for sampling may need more justification. Page 7 line 13-14: the authors could clarify that the survey being used for HIV is the one cited in reference [31] ostensibly from 2013/14, published officially in 2017 – are the dates for data collection therefore correct, and were these results on CHE previously published? Page 7 line 34-37: exclusion of TB patients with <1 month on treatment – would this bias the results, if those patients who drop out early due to, for example, high out-of-pocket costs, are excluded? Page 7 line 38-40: authors should motivate this period of recall, especially given that even in 2014 the majority HIV patients most likely did not visit the facility monthly or even every alternate month, and hence would not have had many outpatient service delivery episodes for which out-of-pocket costs were incurred. In the case that household economic burden was being assessed on a part-year basis, the authors need to explain the choice of the recall period. Page 8 line 20-21: on annualization of costs, the methodological point made is not clear for TB – missing word? Why were 4 visits for HIV/year assumed (this can be linked with reviewer comments earlier)? This issue is also linked with the calculation shown in Table 1 on total outpatient and inpatient annual visits Page 8 line 32-39: the authors should discuss the implication of the two different sources of information used to create proxies of household welfare level (income vs. consumption aggregates)? As we know from decades of household survey-based studies, income data in sub-Saharan Africa with its high levels of informality and under/seasonal employment is a poor self-reported estimator due to the variety of flows, besides the fact that it is often misreported out of respondent caution. Page 10 line 15-20: for purposes of this study, do we know if these episodes of hospitalization were all due to TB or HIV-related conditions? Page 11: Are the authors able to report how many (%) of the TB and HIV patients had both inpatient and outpatient care visits, as a cost estimate for this will be presented later? Page 12-13: Units for Table 2 are missing. Are these in US dollars? For the costs presented in brackets for inpatient care – apparently these are “per single visit” but exceed the value outside the bracket. Could the authors clarify what they intend? [Page 13: As figures are not labelled in the rendered PDF, it was hard to conclusively identify Figure 1.] Page 13 line 42-44: regarding the comment “median cost... seems to be a bit higher for TB patients” – the cost differences between HIV and TB are quite pronounced for higher income / consumption quintiles. Before reading more into this, we recall that the underlying distributions of income vs. cost may vary significantly in the population, and the time period of the observations also vary by 5 years between diseases. The authors may need to comment on this further. General comments (major comments): ----- First, the interpretation of the incidence and intensity of CHE is complicated by the fact that the authors use income and consumption value for HIV and TB-related computations, but with similar thresholds of burden for medical care-related costs. The authors should provide some discussion of the possible bias in comparing the resulting CHE statistics given these differences in
--	---

the computation between the disease driven by the differences in the underlying datasets. They may refer to the chapter on measurement of these burdens and the differences in use of income or consumption in the "Tracking Universal Health Coverage: 2017 Global Monitoring Report" (WHO, 2017). Use of income or consumption has different implications on the estimation of the underlying socioeconomic welfare standard of the household: stated simply, consumption variables could overstate the condition of the household because of use of dissaving/borrowing to smooth consumption over time; whereas using income could understate the condition because of the same effect. Hence, these different choices of the underlying variable used to measure or proxy for socioeconomic welfare can complicate (understate and overstate respectively) when trying to compare the incidence of CHE across population groups stratified on these variables (Page 14 lines 24-27, for example). The use of concentration indices is therefore a better mechanism to study the inequality in the incidence of OOP costs. The authors also need to specify what threshold was used for the determinants of CHE analysis, and whether this was performed using the Xu or Wagstaff method.

Specific comments:

Page 16 Table 4: the result on insurance and increased adjusted odds of facing CHE was surprising. How do the authors account for the lack of protection offered by CBHI? Or does CBHI status proxy for an underlying socioeconomic category (poor/poorest)? The authors could provide tests of association for this.

Second, in the discussion, the authors assert that the cost to patient for HIV care was \$23/visit, whereas this is based on an assumption that there were 4 visits per year. The comparisons between HIV and TB made in the discussion seem to disregard the service delivery realities and trends in sub-Saharan Africa even for their data collection periods for this study (mid 2014 for HIV and late 2019 for TB). The comparisons are also problematic because of methodological issues discussed above. On Page 17 lines 28-31, they make a comparison of the HIV result to a prior study in Ethiopia of the financial burden of TB, and assert as explanation differences in HIV service delivery that are not based on the difference in the care pathway and service delivery orientation between the two diseases. While TB and HIV are often comorbidities, the service delivery patterns and implications for patient care are quite different. The authors may wish to consider that visits to healthcare providers and diagnostic testing during the intensive phase of directly observed short-course therapy for TB imply a very onerous set of direct and indirect costs for patients (e.g., for travel). These costs are usually incident over a limited duration of time up to potential treatment completion and possible sputum conversion. Relapse or retreatment cases and MDR-TB cases may face more intensive and longer duration costs. However, HIV antiretroviral therapy (ART) is a long-course, potentially lifelong course of medication and intermittent diagnostic testing. ART patients are expected, as is the case in Ethiopia, to pick-up their medication occasionally, and see a clinical provider on a quarterly or semi-annual basis, dependent on their stability and success on therapy. The implications for household spending on out-of-pocket costs are very different which will affect interpretation of the results presented. The authors do not revisit their study design by considering these different medical care experiences of the two cohorts of patients that they examine in a

	single study, and what this might mean in instrument design or consideration of the types of recall periods and costs to be included. For HIV: the international comparisons in the discussion section are not well-made; they should be preceded by some comments on the comparability of the cases in terms of time period, health system setting, and country income level. Without some context on how HIV service delivery is handled across the settings from which additional cost estimates are produced for comparison, it is hard to judge whether the comparisons are meaningful or what the basis of the differences/similarities may be. This is important for policy conclusions and search for ways to reduce the financial burden for HIV and TB patients in low and lower-middle income countries. Finally, they contend that “lower cost in our study” (Page 17 line 55-58) was due to “wider scale up of ART in recent years”, whereas the data used in their HIV estimates are of 2014 vintage; when ART coverage in Ethiopia was more modest (about 52-54% among all people living with HIV). They may investigate alternative explanations as well. They also do not provide much of a hypothesis for the positive association of income and HIV costs to the patient. While a finding of preponderance of CHE incidence among the poor is a standard finding, they do not discuss whether there were any impoverishment effects, which is not a part of their analysis overall for either disease. For TB: the authors relate the share of direct costs (46% on average of the total) and the share of non-medical/indirect costs (20% of total) to a conclusion that “diagnostic delays” should be minimized. They use an acronym ‘FRP’ which is not expanded (Page 19 line 34). These conclusions are not well-motivated nor useful when compared to the intentions of the study or the Ethiopian TB program. Separately for TB and the comparisons to other studies, again, comparisons and interpretations of differences of this study with previous studies are haphazardly drawn and summarily dismissed as arising due to “cost estimation method, study setting, health system and socioeconomic differences”, which calls into question the very basis of drawing such international comparisons in the first place. The authors may provide some more nuanced understanding of the difference from previous studies within Ethiopia, accounting for the difference in time period of data collection and shifts in socioeconomic issues as well as differences in TB care design across time. Alluding to differences in “food supplementation expenses” (quite large differences) and valuation of indirect costs is not sufficient when they affect the interpretation of the current results and their validity. There have been relatively fewer changes to the design of first-line TB care and diagnosis, though clinical diagnosis methods have improved with use of molecular diagnosis. These issues of inadequate comparison and interpretation with other studies from Ethiopia follow for analysis of differences in the disaggregated cost results (direct costs and indirect costs). On page 20 (lines 10-17) they assert that CBHI is “marginally protective” which runs counter to the results in Table 4 (where the aOR for CHE of those with insurance is 2.7 compared to no insurance) and text earlier in the paper. This is a fairly glaring error. Third, in the study limitations/strengths, the authors discuss the overestimation of costs when aggregating annually. This could have been managed by some sensitivity analysis of different
--	---

	aggregation-related assumptions (related to the standard of care and/or average patient experiences from other sources). Bias introduced due to the geographical specificity of the sample were not fully explained in the methodological section, and hence the limitation is hard to value when presented late in the paper. Recommendations in the paper related to increasing financial protection are weak, and not related to health financing issues arising in Ethiopia today related to deepening of CBHI or social health insurance, or in terms of models of differentiated service delivery for HIV or other methods to care for TB patients or offer them financial assistance during the intensive phase of directly observed short-course therapy, first or second line. ----- Final minor comments: Additional to these issues, there were numerous syntactical issues which may need to be addressed in a major revision.
--	--

VERSION 1 – AUTHOR RESPONSE

Reviewer 1

1. Please state any competing interests or state 'None declared': None

Reply: We have now modified the sentence on page number 22, line 453 as, “Competing Interests: None declared”.

2. Thank you for this interesting article. This article is well-written and scientifically sound. However, the reference style and in-text citations format of the manuscript is not in accordance to the journal's instruction for authors. Please revise - go through an already published article to see how it should be structured.

Reply: Thank you, the reference style and in text citations are now updated according to the journal's instructions.

Reviewer 2

3. The stated rationale and methods for the study could be refined. The case for catastrophic health expenditures (CHE) is made very strongly, perhaps too strongly, and uses data from other settings than Ethiopia. The range of costs “per TB episode” stated is very high and from a systematic review with studies conducted prior to 2005; and the relevance to the situation in Ethiopia in late 2019 could be more carefully posed, where TB care is an “exempted service” and first- and second-line drugs are available for free based on external grant funding.

Reply: We accept that the rationale required additional inputs and that the case for CHE may be a bit overemphasized with the wide ranges reported. We have now updated this with latest (2015) figures from low-income countries. Therefore, we added the sentence on page number 4 and line number 80-83 “A systematic review showed that individuals in low income countries spend a mean direct cost of \$155 per drug susceptible TB (DS) and \$406 per drug-resistant TB (DR-TB). The productivity losses were 2-3 times higher than the direct costs of DS and DR-TB cases, respectively”. We need to highlight that also, according to the recent WHO global TB report (2019), TB affected households were facing high costs despite free TB services. Such costs accounted for a large proportion of household income and further pushed patients to catastrophic health expenses. Though TB care is among the “exempted services” in Ethiopia, patients pay for the services that are not in the package

of basic TB care (e.g. for X-ray, chemistry tests), for non-medical expenses, and for services in private care settings.

4. Page 6 line 36-37: clarify “sample of 105 HIV associations”. What type of sampling units are these? With PPS sampling, how did the authors ensure that areas with important characteristics but smaller “HIV associations” were included?

Reply: Even though the population of interest for the HIV survey was all adults living with HIV in the country, it was not feasible to use this population as the sample frame because several people in Ethiopia do not know they are HIV-infected, and there is no database of those who know from which to establish the sampling frame. In view of this, the study used PLHIV associations under Network of HIV Positives in Ethiopia (NEP+) as the sampling unit to select the study participants -PLHIV. We added a description for the sampling unit on page 6 and line number 121-3, “For HIV, PLHIV associations were used as a sampling frame to select HIV participants, as there is no national registry of PLHIV. The association operates in big cities and their members were mainly urban residents”. In this study, in the first stage, PLHIV associations were selected according to the number of associations in each region. The same number of units within the selected associations (i.e. whether they are bigger or smaller associations) had been sampled. We modified the sentence in page number 6 line number 126-7 “In stage two, 40 HIV members from each sampled association, in total 4,171, were randomly selected and interviewed”.

5. Page 6 line 48-49, for sampling, the point estimate of the share of households that may incur CHE in the top income quartile is from a study in Peru. Its suitability to Ethiopia as a basis for sampling may need more justification.

Reply: The comparison with Peru can be justified because Peru has an epidemiology close to that of Ethiopia and is one of the high TB burden countries with the same proportion of TB/HIV co-infections as in Ethiopia.

6. Page 7 line 13-14: the authors could clarify that the survey being used for HIV is the one cited in reference [31] ostensibly from 2013/14, published officially in 2017 – are the dates for data collection therefore correct, and were these results on CHE previously published?

Thank you for your comment. The HIV data collection took place in 2016 and was published in 2017. However, in the main survey the HIV related expenditure was deflated for the year 2014, to make the report consistent with the national health account survey 2013/14, as the HIV survey is part of the NHA survey even though it takes place at the end of 2016. This is the main reason that caused the confusion. We have now made the required correction in both the main manuscript and the reference at page 6 and line number 113. The sentence now reads, “Data for HIV were collected from mid-September to mid-October 2016”. Similarly, the reference was modified accordingly. In addition, as the data collection period for HIV was in late 2016, we have made the required correction in the HIV result (i.e. Table 2-4 and Figure 1-3).

We also would like to clarify that the survey mainly reported the health service utilization patterns and expenditures on health among people living with HIV. However, in our study we further analysed the data to report not only the direct costs, but also the indirect costs, catastrophic health expenditure and determinant factors associated with seeking HIV care that are not addressed in the main survey.

7. Page 7 line 34-37: exclusion of TB patients with <1 month on treatment – would this bias the results, if those patients who drop out early due to, for example, high out-of-pocket costs, are excluded?

This is a good point, and we may lose patients that drop out early. However, the approach we used for inclusion is in line with the WHO guide for TB patient cost surveys. The WHO recommendation is to invite all consecutive TB and DR-TB patients on treatment follow up at sampled facility for a minimum of 2 weeks. This timeline was also chosen to show the need to have a good balance between recall bias and cost experience. We included the following sentence on page 7 and line number 151-52 “This timing is selected because patients’ easily recall their pathway to TB care and would acquire adequate cost experience”.

8. Page 7 line 38-40: authors should motivate this period of recall, especially given that even in 2014 the majority HIV patients most likely did not visit the facility monthly or even every alternate month, and hence would not have had many outpatient service delivery episodes for which out-of-pocket costs were incurred. In the case that household economic burden was being assessed on a part-year basis, the authors need to explain the choice of the recall period.

Reply: We appreciate the reviewer’s comment and agree that the recall period was very short. Nonetheless, during that time (2016), patients were required to visit health facilities regularly (i.e. on monthly basis) if they had adherence problems, were pregnant, or experienced treatment failure etc. Similarly, patients visited health facilities at any time, if they required treatment of opportunistic infections or other healthcare needs, e.g. side effects of medication. Otherwise, most will visit every other month or on quarterly basis. In addition to this, the sample size used was sufficient with an adequate number of outpatient service attendants participating within the specified period. There is also a trade-off between capturing all costs with a larger recall period (which would have high recall bias) and a shorter recall period (which would reduce recall bias).

9. Page 8 line 20-21: on annualization of costs, the methodological point made is not clear for TB–missing word? Why were 4 visits for HIV/year assumed (this can be linked with reviewer comments earlier)? This issue is also linked with the calculation shown in Table 1 on total outpatient and inpatient annual visits.

Reply: We agree that this needs more precision and that the sentences on page 8 and line number 170-171 have now been revised as “In order to annualize the cost, an average of four HIV outpatient visits and single TB episode outpatient visits was considered”. From the HIV household survey, we estimated the health care utilization rate among all participants surveyed in the past one month. This was converted to annual basis by assuming 4 visits per patient per year since this is supported by previous study findings from Ethiopia (1).

10. Page 10 line 15-20: for purposes of this study, do we know if these episodes of hospitalization were all due to TB or HIV-related conditions?

Reply: The hospitalizations were mainly related to TB conditions for TB patients and related to HIV co-infection like tuberculosis, respiratory and diarrheal diseases for HIV patients.

11. Page 11: Are the authors able to report how many (%) of the TB and HIV patients had both inpatient and outpatient care visits, as a cost estimate for this will be presented later?

Reply: Yes, we have now modified the sentence on Page 10, line number 214-15; it now reads: “About 22% and 6% of HIV and TB patients had both outpatient and inpatient care, respectively”.

12. Page 12-13: Units for Table 2 are missing. Are these in US dollars? For the costs presented in brackets for inpatient care – apparently these are “per single visit” but exceed the value outside the bracket. Could the authors clarify what they intend?

Reply: Yes, these are all in US\$. We have now added the appropriate unit under the headings of tables 2 and 3 “expressed in \$” and after the heading “Subcategory (\$)”. In this study, we present both mean and median values in the analysis. The mean was reported because it provides valuable aggregate information. Further, we supplemented with median values due to the skewed nature of the data. In this study, the data is more right skewed and the standard deviation for some of the variables is greater than the mean showing the data as more spread out. Therefore, we modified the sentence in page 9 and line number 187-89. It now reads, “The data was summarized using mean with standard deviation (SD) or median with interquartile range (IQR) due to skewed distribution”.

13. [Page 13: As figures are not labelled in the rendered PDF, it was hard to conclusively identify Figure 1.]

Reply: Thank you. We have now resubmitted the figures with labels again and we hope that this fixes the problem.

14. Page 13 line 42-44: regarding the comment “median cost... seems to be a bit higher for TB patients” – the cost differences between HIV and TB are quite pronounced for higher income /consumption quintiles. Before reading more into this, we recall that the underlying distributions of income vs. cost may vary significantly in the population, and the time period of the observations also vary by 5 years between diseases. The authors may need to comment on this further.

Reply: Please see comments 6 and 12 above.

15. a) The authors have added usefully to the literature on the economic burden of accessing care for HIV and TB in sub-Saharan Africa. There are some matters that could benefit from a revision. First, the interpretation of the incidence and intensity of CHE is complicated by the fact that the authors use income and consumption value for HIV and TB-related computations, but with similar thresholds of burden for medical care-related costs. The authors should provide some discussion of the possible bias in comparing the resulting CHE statistics given these differences in the computation between the disease driven by the differences in the underlying datasets. They may refer to the chapter on measurement of these burdens and the differences in use of income or consumption in the “Tracking Universal Health Coverage: 2017 Global Monitoring Report” (WHO, 2017). Use of income or consumption has different implications on the estimation of the underlying socioeconomic welfare standard of the household: stated simply, consumption variables could overstate the condition of the household because of use of dissaving/borrowing to smooth consumption over time; whereas using income could understate the condition because of the same effect. Hence, these different choices of the underlying variable used to measure or proxy for socioeconomic welfare can complicate (understate and overstate respectively) when trying to compare the incidence of CHE across population groups stratified on these variables (Page 14 lines 24-27, for example). The use of concentration indices is therefore a better mechanism to study the inequality in the incidence of OOP costs. The authors also need to specify what threshold was used for the determinants of CHE analysis, and whether this was performed using the Xu or Wagstaff method.

b) Page 8 line 32-39: the authors should discuss the implication of the two different sources of information used to create proxies of household welfare level (income vs. consumption aggregates)? As we know from decades of household survey-based studies, income data in sub-Saharan Africa with its high levels of informality and under/seasonal employment is a poor self-reported estimator due to the variety of flows, besides the fact that it is often misreported out of respondent caution.

Reply: We agree that consumption is a better proxy for household welfare, and that comparing income and consumption is a constraint. However, we only had income data for HIV and since we conducted the TB survey ourselves, we chose to focus on consumption for the exact same reasons

as pointed out here. In the HIV survey, consumption data were not available, as the majority of the survey population provides complete income data and resides in urban areas. Hence, we used the reported income as a proxy for living standard measure despite its limitation. We have therefore included the sentence in the discussion section Page 20, line numbers 400-406 to be clearer about these limitations: “The comparison of incidence and intensity of CHE for TB and HIV is also complicated by the use of income for HIV and consumption for TB-related computations. In developing countries, income is poor self-reported estimator of welfare due to more common informal employment, seasonal agricultural activities and widespread reluctance to disclose income. Therefore, income could understate the welfare of the household, whereas using consumption may overstate the condition of the household because of the use of dissaving/borrowing to smooth consumption over time”.

We also had specified the threshold level used for the determinant analysis of CHE. The sentence in page 15 and line number 306-7 reads now: “Multivariate logistic regression model of determinants of CHE for TB and HIV care at a 10% threshold of household income/consumption (Ethiopia)”.

16. Page 16 Table 4: the result on insurance and increased adjusted odds of facing CHE was surprising. How do the authors account for the lack of protection offered by CBHI? Or does CBHI status proxy for an underlying socioeconomic category (poor/poorest)? The authors could provide tests of association for this.

Reply: We thank the reviewer for this comment. This was a typo error for the insurance variable comparison group and we have now edited table 4 accordingly. In our study, the un-insured had increased odds of CHE. The sentence in page 15 line number 304-5 reads: “Households with a health insurance scheme have protection from CHE (aOR 2.7; 95%CI 1.06–6.73)”.

17. Second, in the discussion, the authors assert that the cost to patient for HIV care was \$23/visit, whereas this is based on an assumption that there were 4 visits per year. The comparisons between HIV and TB made in the discussion seem to disregard the service delivery realities and trends in sub-Saharan Africa even for their data collection periods for this study (mid 2014 for HIV and late 2019 for TB). The comparisons are also problematic because of methodological issues discussed above. On Page 17 lines 28-31, they make a comparison of the HIV result to a prior study in Ethiopia of the financial burden of TB, and assert as explanation differences in HIV service delivery that are not based on the difference in the care pathway and service delivery orientation between the two diseases. While TB and HIV are often comorbidities, the service delivery patterns and implications for patient care are quite different. The authors may wish to consider that visits to healthcare providers and diagnostic testing during the intensive phase of directly observed short-course therapy for TB imply a very onerous set of direct and indirect costs for patients (e.g., for travel). These costs are usually incident over a limited duration of time up to potential treatment completion and possible sputum conversion. Relapse or retreatment cases and MDR-TB cases may face more intensive and longer duration costs. However, HIV antiretroviral therapy (ART) is a long-course, potentially lifelong course of medication and intermittent diagnostic testing. ART patients are expected, as is the case in Ethiopia, to pick-up their medication occasionally, and see a clinical provider on a quarterly or semi-annual basis, dependent on their stability and success on therapy. The implications for household spending on out-of-pocket costs are very different which will affect interpretation of the results presented. The authors do not revisit their study design by considering these different medical care experiences of the two cohorts of patients that they examine in a single study, and what this might mean in instrument design or consideration of the types of recall periods and costs to be included.

Reply: This is an important point with no simple answer. Therefore, we have tried to reflect a bit more on this in the revised manuscript. The data collection time for HIV and the choice of 4 visits per year are explained under comment 9 to reviewer #2 above. In addition, in the discussion section, we have modified the sentence on page 19-20 and line number 393-400: now reads, “The TB related CHE was

higher than the reported rate for HIV. One important reason for this is the difference in the duration of treatment and intervention characteristics between the two diseases. TB patients experience a very onerous set of direct and indirect costs during diagnostic and intensive phase of directly observed short course therapy (i.e. more intense for retreatment, extra-pulmonary and DR-TB cases). After treatment completion and possible sputum conversion, TB patients are less likely to face additional costs. Even though the annual cost of HIV care is lower, PLHIV faces these costs over its lifetime because HIV infection is a chronic disease that needs lifelong treatment”.

18. For HIV: the international comparisons in the discussion section are not well-made; they should be preceded by some comments on the comparability of the cases in terms of time period, health system setting, and country income level. Without some context on how HIV service delivery is handled across the settings from which additional cost estimates are produced for comparison, it is hard to judge whether the comparisons are meaningful or what the basis of the differences/similarities may be. This is important for policy conclusions and search for ways to reduce the financial burden for HIV and TB patients in low and lower-middle income countries. Finally, they contend that “lower cost in our study” (Page 17 line 55-58) was due to “wider scale up of ART in recent years”, whereas the data used in their HIV estimates are of 2014 vintage; when ART coverage in Ethiopia was more modest (about 52-54% among all people living with HIV). They may investigate alternative explanations as well. They also do not provide much of a hypothesis for the positive association of income and HIV costs to the patient. While a finding of preponderance of CHE incidence among the poor is a standard finding, they do not discuss whether there were any impoverishment effects, which is not a part of their analysis overall for either disease.

Reply: As the dataset used for HIV was in late 2016, which validates the sentence “lower cost in our study” (Page 17 line 345-6) was due to “wider scale up of ART in recent years”. This was also supported by the fact that the country introduced test and treat all strategy during this period. We did not discuss the level of impoverishment, which was beyond the scope of this study.

19. For TB: the authors relate the share of direct costs (46% on average of the total) and the share of non-medical/indirect costs (20% of total) to a conclusion that “diagnostic delays” should be minimized. They use an acronym ‘FRP’ which is not expanded (Page 19 line 34). These conclusions are not well-motivated nor useful when compared to the intentions of the study or the Ethiopian TB program. Separately for TB and the comparisons to other studies, again, comparisons and interpretations of differences of this study with previous studies are haphazardly drawn and summarily dismissed as arising due to “cost estimation method, study setting, health system and socioeconomic differences”, which calls into question the very basis of drawing such international comparisons in the first place. The authors may provide some more nuanced understanding of the difference from previous studies within Ethiopia, accounting for the difference in time period of data collection and shifts in socioeconomic issues as well as differences in TB care design across time. Alluding to differences in “food supplementation expenses” (quite large differences) and valuation of indirect costs is not sufficient when they affect the interpretation of the current results and their validity. There have been relatively fewer changes to the design of first-line TB care and diagnosis, though clinical diagnosis methods have improved with use of molecular diagnosis. These issues of inadequate comparison and interpretation with other studies from Ethiopia follow for analysis of differences in the disaggregated cost results (direct costs and indirect costs). On page 20 (lines 10-17) they assert that CBHI is “marginally protective” which runs counter to the results in Table 4 (where the aOR for CHE of those with insurance is 2.7 compared to no insurance) and text earlier in the paper. This is a fairly glaring error.

Reply: Thank you for your comment. Related to the conclusion that diagnostic delays should be minimized because of the large share of direct costs, we agree and modified the sentence in page 19, line 386-7. Now this reads: “Therefore, ensuring the expansion of TB service package through the

effective integration of a health insurance scheme and decentralization of services can reduce the direct costs.”

The FRP acronym is expanded within the first usage in page 4 line number 76 “... financial risk protection (FRP)).”

We had tried to address the comment related “...understanding of the difference from previous studies within Ethiopia...” by capturing the main differences in terms of seeking care, type of TB attending health facilities, and delay in seeking care. Accordingly, the sentence in page 18 and line number 364-68. It now reads: “The variation in cost from previous studies arises from high expenditure on nutritional supplements (\$72 vs \$25)[29], lower level of seeking diagnostic care from public health facilities(64% vs 85%), high rates of clinically diagnosed cases (49% vs 24%), diagnostic delays result in longer losses of working days[22], only direct cost captured[47]”.

We have clarified the confusion related to CBHI; please see the clarification given to reviewer # 2 above in comment 16.

20. Third, in the study limitations/strengths, the authors discuss the overestimation of costs when aggregating annually. This could have been managed by some sensitivity analysis of different aggregation-related assumptions (related to the standard of care and/or average patient experiences from other sources). Bias introduced due to the geographical specificity of the sample were not fully explained in the methodological section, and hence the limitation is hard to value when presented late in the paper.

Reply: We have tried to put the geographical specificity of the sample on page 6 and line number 122-3. The sentence now reads, “The association operates in big cities and its members were mainly urban residents”. In addition, “... for TB, data were collected from December 2018 to September 2019 in three zones (i.e. zone 3 of Afar region, and Jimma and Adama special zones of Oromia region)” in page 6 and line 114-116.

21. Recommendations in the paper related to increasing financial protection are weak, and not related to health financing issues arising in Ethiopia today related to deepening of CBHI or social health insurance, or in terms of models of differentiated service delivery for HIV or other methods to care for TB patients or offer them financial assistance during the intensive phase of directly observed short-course therapy, first or second line.

Reply: Thank you. We modified and enriched the recommendation in page 21 and line number 433-436. It now reads, “Second, strategies to ensure social and FRP for HIV and TB affected households are necessary. This requires effective integration of HIV and TB services with health insurance, provision of differentiated care and community-based treatment for stable patients, and enabling access to social protection schemes”.

22. Final minor comments: Additional to these issues, there were numerous syntactical issues which may need to be addressed in a major revision.

Reply: Thank you. We had attempted to correct the syntactical issues accordingly.

References

1. Bikilla A, Jerene D, Robberstad B, Lindtjorn B. Cost estimates of HIV care and treatment with and without anti-retroviral therapy at Arba Minch Hospital in southern Ethiopia. Cost effectiveness and resource allocation : C/E. 2009;7:6.

VERSION 2 – REVIEW

REVIEWER	Arin Dutta Palladium United States
REVIEW RETURNED	26-Mar-2020

GENERAL COMMENTS	I thank the authors for making very helpful changes and responding in detail to the review. A few minor issues could be clarified.  1. Thank you for edits in the sampling methods. On the issue of sampling of HIV associations, Could you please include a short statement whether the share of people who are members of PLHIV associations are representative of the broader population of PLHIV (including non-members), especially when it comes to access to care, level of information, socioeconomic status. This goes to generalizability. 2. On the inclusion/exclusion of types of TB patients (e.g., exclude <1 month), the authors could cite the WHO guide for TB patient cost surveys as justification. The link between the ease of recall and the decision to drop those with <1 month of Tx could be better forged. 3. Related to the HIV period of recall, I appreciate the authors' response. Can they include some small sample or program implementation study that confirmed the majority of HIV patients in the time period of data collection were on more frequent scripting and clinic visit scheduling in Ethiopia? This is also linked to the HIV annualization assumption of 4 visits/year, given the authors attest in the response that "most will visit every other month" (suggesting 6 visits?) 4. I appreciate the response and modifications made by the authors to the difficult questions around use of appropriate measures of socioeconomic status and calculation of CHE, especially around the difficulties of using income and consumption variables. I think the caveats they have now included are satisfactory. While the language now included on the issue of difference in service delivery and patient experience between TB and HIV, and hence the influence on interpretation of OOP/CHE, is welcome, it could be further strengthened. One of the ways to consider strengthening this is to reflect on what the implications are for providing financial protection for TB patients due to the frequency of clinical visits during DOTS intensive phase. 5. I still remain somewhat unconvinced of the link between "lower cost" as found in this study for HIV and "ART scale-up in recent years", in terms of the causal link the writers are asserting. Patients who got immediate offers to start treatment may skip the visits associated with pre-ART, but they were not a significant number in the years where CD4-based rationing of ART was in effect. Even if they started ART immediately, they would be subject to the same 4+ visits per year the authors have assumed. I am not clear on what protective effect is intended from the scale-up occurring during the period in question (2016). I urge the authors to clarify this point, as it is important that we do give credit where it is due. I think avoiding OOP spending for HIV patients may have more to do at this stage with multi-month scripting,
---

	avoidance of unnecessary tests which are not clinically required and reduced clinical visits if stable. 6. Minor quibble, but I urge the authors not to include non-standard abbreviations and acronyms for common concepts like financial risk protection. 7. I thank the authors for making changes to the discussion and making references to social protection schemes inclusive of CBHI more prominent. However, they should provide some caveats to make sure that these recommendations are not perceived as facile. The CBHI scheme has financial challenges and they could recognize that further integration of financial protection for HIV and TB patients, such as paying transport subsidies, may not be possible for the scheme (the services are already free, as are the commodities). What exactly can be done to reduce the OOP burden for things which are more of a broader social protection nature? Can woredas and community organizations provide some support to vulnerable TB patients and people living with HIV who need support in accessing care? Thank you.
--	---

VERSION 2 – AUTHOR RESPONSE

Reviewer 1

1. Please state any competing interests or state 'None declared': None declared

Reply: We have stated competing interests ("None declared") at page 22, line 450.

2. Thank you for edits in the sampling methods. On the issue of sampling of HIV associations, Could you please include a short statement whether the share of people who are members of PLHIV associations are representative of the broader population of PLHIV (including non-members), especially when it comes to access to care, level of information, socioeconomic status. This goes to generalizability.

Reply: We appreciate the reviewer's comment. However as pointed out in the previous version of this paper: "The association operates in major cities in all regions and its members were primarily residents in urban areas" (method section, page 6 and lines 122-23). Therefore, given the fact that the association operates in all regions and highest HIV prevalence areas, it is not representative of rural and non-member persons living with HIV. We have addressed this limitation on page 21 line 424-25, which now reads "...Similarly, the PLHIV associations operate in high HIV prevalence urban areas, members of these associations may not be representative of both rural and non-members."

3. On the inclusion/exclusion of types of TB patients (e.g., exclude <1 month), the authors could cite the WHO guide for TB patient cost surveys as justification. The link between the ease of recall and the decision to drop those with <1 month of Tx could be better forged.

Reply: We agree with the reviewer and cited the WHO Guide on page 7 and lines 151-52. The revised sentence now reads: "This schedule of interview is based on WHO's recommendation regarding the cost survey of TB patients [1,2]".

4. Related to the HIV period of recall, I appreciate the authors' response. Can they include some small sample or program implementation study that confirmed the majority of HIV patients in the time period of data collection were on more frequent scripting and clinic visit scheduling in Ethiopia? This is also linked to the HIV annualization assumption of 4 visits/year, given the authors attest in the response that "most will visit every other month" (suggesting 6 visits?)

Reply: Thank you. The frequency of HIV visit per year was estimated from the main survey. We first estimated the proportion of individuals seeking care among all PLHIV interviewees over the last month and then annualized this rate to get around 4 visits per patient per year. It is also supported by the 2014 National Guidelines for Comprehensive HIV Prevention, Care and Treatment, Ethiopia where it recommends PLHIV visit to health facilities more often if they have adherence issues, are in early phase of ART initiations, are pregnant, experienced treatment failure or other similar clinical conditions. Additionally, patients visit health facilities at any time when they need treatment of opportunistic diseases or other healthcare needs, e.g. medication side effects. Otherwise, patients with stable HIV condition will return every twelve weeks [3]. This statement is also backed by earlier Ethiopian studies [4].

In summary, to elaborate more on HIV visits per year, we added the sentence on page 8 lines 171-4. The sentence now reads: "The frequency of outpatient HIV visits per year was extrapolated on the basis of per capita health care visits an individual made over the last one month among all PLHIV interviewed. This proportion was annualized to an approximately 4 visits per patient and year."

5. I appreciate the response and modifications made by the authors to the difficult questions around use of appropriate measures of socioeconomic status and calculation of CHE, especially around the difficulties of using income and consumption variables. I think the caveats they have now included are satisfactory. While the language now included on the issue of difference in service delivery and patient experience between TB and HIV, and hence the influence on interpretation of OOP/CHE, is welcome, it could be further strengthened. One of the ways to consider strengthening this is to reflect on what the implications are for providing financial protection for TB patients due to the frequency of clinical visits during DOTS intensive phase.

Reply: We appreciate the reviewer's suggestion. We addressed the comment concerning the provision of financial protection for frequency of visit during intensive phase together with reviewer's comment # 8 below and suggested relevant recommendations (please see response # 8).

Furthermore, to emphasize the sentence on the difference in service delivery and patient experience between TB and HIV, we made the following revision on page 19 lines 395-96: "The main reasons for

this are the differences in treatment duration, follow-up frequency and care access between HIV and TB.”

6. I still remain somewhat unconvinced of the link between "lower cost" as found in this study for HIV and "ART scale-up in recent years", in terms of the causal link the writers are asserting. Patients who got immediate offers to start treatment may skip the visits associated with pre-ART, but they were not a significant number in the years where CD4-based rationing of ART was in effect. Even if they started ART immediately, they would be subject to the same 4+ visits per year the authors have assumed. I am not clear on what protective effect is intended from the scale-up occurring during the period in question (2016). I urge the authors to clarify this point, as it is important that we do give credit where it is due. I think avoiding OOP spending for HIV patients may have more to do at this stage with multi-month scripting, avoidance of unnecessary tests which are not clinically required and reduced clinical visits if stable.

Reply: Thanks for your suggestion. The previous explanation in the paper “the lower cost in our study may be due to the wider scale up of ART in recent years, which minimizes costs of pre-ART treatment and advanced HIV infection management” is mainly intended to provide the rationale on why the HIV related productivity loss in our study is lower than the previous study. We have made the corresponding revision on page 17 lines 345-47: “The difference in productivity losses with previous Ethiopian study could be attributed to a more centralized provision of HIV services and backlog of patients with advanced HIV diseases, unlike this study.”

7. Minor quibble, but I urge the authors not to include non-standard abbreviations and acronyms for common concepts like financial risk protection.

Reply: Thank you. The abbreviation referring to financial risk protection (i.e. FRP) in the manuscript was replaced with full sentences.

8. I thank the authors for making changes to the discussion and making references to social protection schemes inclusive of CBHI more prominent. However, they should provide some caveats to make sure that these recommendations are not perceived as facile. The CBHI scheme has financial challenges and they could recognize that further integration of financial protection for HIV and TB patients, such as paying transport subsidies, may not be possible for the scheme (the services are already free, as are the commodities). What exactly can be done to reduce the OOP burden for things which are more of a broader social protection nature? Can woredas and community organizations provide some support to vulnerable TB patients and people living with HIV who need support in accessing care?

Reply: We thank the reviewer for this comment. We still believe that the integration with CBHI is necessary because only a small package of TB and HIV services have been provided free of charge. Patients pay for ancillary medications, basic laboratory testing, imaging, pre-diagnostic services, adverse event monitoring, etc. Hence, expansion of insurance coverage for TB-related services and HIV related comorbidities through integration of the service with health insurance is important to

reduce medical costs. Nevertheless, not all costs associated with seeking HIV and TB care are alleviated by CBHI. We also advocate local health care models that bring the services closer to patients, including community-based treatment and appointment spacing for stable patients that reduce some of the travel costs.

Accordingly, the recommendation on page 21 lines 433-39 had been updated. The sentence now reads: “First, more patient-centered care with effective diagnostics, appointment spacing for stable patients and community-based treatment are required to improve the delivery of HIV and TB services. Second, strategies are required to ensure social and financial risk protection for the households affected by HIV and TB. This requires effective integration of HIV and TB services with social and financial protection schemes, including the provision of travel vouchers, nutritional support and paid sick leaves through multi-sectoral collaboration.”

Reference

- 1, Ukwaja KN, Alobu I, Igwenyi C, et al. The High Cost of Free Tuberculosis Services: Patient and Household Costs Associated with Tuberculosis Care in Ebonyi State, Nigeria. PLOS ONE. 2013;8(8):e73134 DOI: 10.1371/journal.pone.0073134.
- 2, WHO. WHO Global TB Programme. Protocol for survey to determine direct and indirect costs due to TB and to estimate proportion of TB-affected households experiencing catastrophic costs. November 2015.
- 3, Federal Democratic Republic of Ethiopia, Ministry of Health. National guidelines for comprehensive HIV prevention, care and treatment. 2014.
- 4, Bikilla A, Jerene D, Robberstad B, Lindtjorn B. Cost estimates of HIV care and treatment with and without anti-retroviral therapy at Arba Minch Hospital in southern Ethiopia. Cost effectiveness and resource allocation : C/E. 2009;7:6.

VERSION 3 – REVIEW

REVIEWER	Arin Dutta Palladium, Americas Health Practice
REVIEW RETURNED	21-Apr-2020

GENERAL COMMENTS	For the recent re-submission, I have reviewed the authors' changes using the tracked edits version and am satisfied they have made a good faith attempt to address all the changes suggested in the second review. Specifically, additional detail on what patients may be still expected to pay in Ethiopia for care notwithstanding the "exempted service" nature of TB and HIV treatment is useful in the background section as this goes to later suggestions of what might be protective policy changes. Edits to sampling, data collection and assumptions sections were also noted and are satisfactory, especially when linked with later edits in the limitations section. The substantive changes in this round relate to interpretations and study limitations, especially when comparing findings to previous analyses in Ethiopia and elsewhere. The change to attribution of the reduced productivity losses for HIV patients in this study
--

	compared to previous studies is now ascribed to "more centralized provision of HIV services" and "backlog of patients with advanced HIV diseases". The meaning of this sentence should be further clarified in syntactical edit, as I think the authors mean to say that the previous study was conducted when those conditions applied, vs. the current study. The change of attribution of the potential factor is welcome, vs. the "major scale up of HIV services" as a possible protective factor (in the previous draft). However, the authors should consider clarifying this in the final version in terms of why centralized provision is less protective, when working with the editor. Finally, the section on policy implications, just before the "Conclusion" section, has been usefully edited and improved solutions are offered for the problem of inadequate financial protection. However, when reading this from a standpoint of logical flow, it seems that these policy implications may be better placed in the "conclusions" section rather than in "strengths and limitations". This may be further discussed with an editor. Overall, I thank the authors for their attention to the reviewers' comments and their patience with the process, and for this timely and thoughtful study.
--	--